# Compositional World Models with Interpretable Abstractions

## Abstract

We present a modular and compositional approach to learning human-aligned world models via state-action hierarchies. Our approach is inspired by sensory-motor hierarchies in the mammalian brain. We model complex state transition dynamics as a sequence of simpler dynamics, which in turn can be modeled using even simpler dynamics, and so on, endowing the approach with rich compositionality. We introduce Composer, a practical method for learning complex world models that leverages hypernetworks and abstract states for generating lower-level transition functions on-the-fly. We first show that state abstractions in Composer emerge naturally in simple environments as a consequence of training. Incorporating a variant of contrastive learning allows Composer to scale to more complex environments while ensuring that the learned abstractions are human aligned. Additionally, learning a higher-level transition function between learned abstract states leads to a hierarchy of transition functions for modeling complex dynamics. We apply Composer to compositional navigation problems and show its capability for rapid planning and transfer to novel scenarios. In both traditional grid-world navigation problems as well as in the more complex Habitat vision-based navigation domain, a Composer-based agent learns to model the state-action dynamics within and between different rooms using a hierarchy of transition functions and leverage this hierarchy for efficient downstream planning. Our results suggest that Composer offers a promising framework for learning the complex dynamics of real-world environments using a compositional and interpretable approach.

## 1 Introduction

Composing existing skills and knowledge to creatively generate solutions for new and complex problems is a fundamental attribute of human intelligence. Advances in generative AI and large language models are beginning to demonstrate attributes of human-like intelligence but fail at simple tasks like multiplying a few small numbers (Bubeck et al. (2023); Bender et al. (2021); Schmidhuber (1991a)) that rely on application of compositional knowledge and reasoning. Similar observations hold true for traditional reinforcement learning (RL) and embodied AI agents (Lake et al. (2016)). Recent developments in hierarchical reinforcement learning, supported by novel architectures, have resolved several such challenges in the field (Hafner et al., 2022; Levy et al., 2017; Kulkarni et al., 2016). However, a significant gap remains in effectively utilizing structured architectures to exploit compositionality and enable rapid transfer of dynamics and skills. Additionally, while Vision Language Models have made labeling visual data faster and more affordable (Radford et al., 2021; Deitke et al., 2024; Liu et al., 2024), there is still limited research on how these labels can enhance compositional learning in embodied agents.

This prompts a key question: What fundamental computational principles in biological neural networks enable compositionality for solving novel problems? To rigorously answer this question, we look towards recent advances in computational neuroscience. Predictive coding theories have consistently garnered increasing attention as computational models of how the brain perceives and acts in the real world (Rao & Ballard (1999); Friston & Kiebel (2009); Keller & Mrsic-Flogel (2018); Jiang & Rao (2022b)). In predictive coding, different areas of the neocortex together implement a hierarchical generative model of the world. Feedback connections from a higher to a lower level predict lower-level responses, and the prediction errors propagate via feedforward connections to update higher-level estimates. While the original formulation of predictive coding ignored actions,

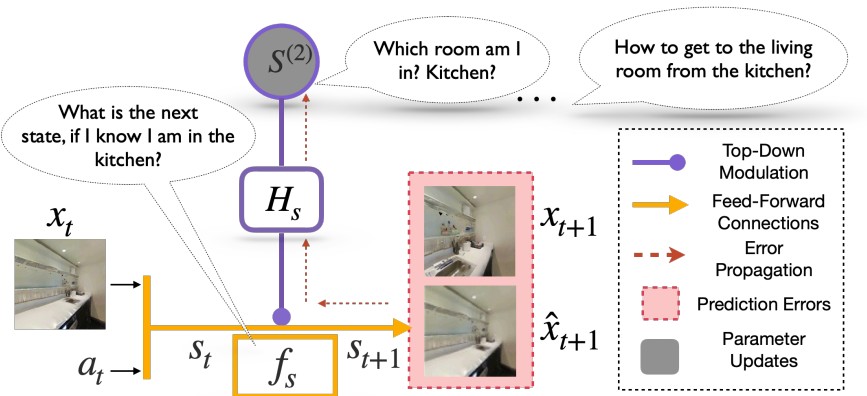

Figure 1: **Learning and Inferring Hierarchical Dynamics**. A learnable higher-level latent state $S^{(2)}$ generates, using a hypernet $H_s$, a lower-level transition function $f_s$ mapping current input $x_t$ and lower-level action $a_t$ to next input. Here, the input is an image (taken from the Habitat environment). Composer uses the sequence of prediction errors between the model prediction and the true input to update and infer in real time the higher-level latent state $S^{(2)}$ (here, representing an estimate for the current room). Complex dynamics are abstracted in an unsupervised manner in a sequence of simpler dynamics which are reused to model dynamics in other problems. Additionally, such abstractions allow hierarchical planning such as navigating between rooms using abstract actions rather than primitive actions, resulting in significant savings.

recent attempts towards neo-cortical modeling integrate actions for learning and modeling the dynamics of environments via state-action hierarchies (Rao et al., 2023; Rao, 2024). Parallel work on hippocampal activity in navigating mice brains have made progress in understanding computationally complex domains such as transfer learning, and hierarchical planning in the cortex (Botvinick et al., 2009; Merel et al., 2019). Grid cells in the entorhinal cortex simulate spatial reference frames that help breakdown a problem into simpler, reusable components. Further, graph schemas, implemented in the hippocampus has shown evidence for compositional learning ( Moser et al. (2008); Guntupalli et al. (2023); Whittington et al. (2021)). The key insight we consistently observe from research in neuroscience is that cortical circuits break down a problem into simple sub-components and solve them via modulated transition dynamics (a.k.a firing patterns) specific to the problem.

Motivated by these insights, we develop in Section 2 the Composer algorithm for learning a hierarchy of transition functions and state abstractions. Composer uses only random trajectory rollouts of an agent and their prediction errors to naturally learn abstractions (Fig. 1). We argue that unlike traditional state and action abstractions, Composer learns to abstract the transition dynamics of environments and reuses them in similar scenarios. We confirm our hypothesis with experiments and present them here. In Section 3 we build on the preliminary insights from the previous section and scale Composer to more complex environments with interpretable notion of abstractions. Finally, in Section 4 we present results for interesting applications with Composer, like hierarchical planning and novel scene generation. Details including additional results, code snippets and derivations are presented in the supplementary section. To summarize, the main contributions of this paper are:

1. A novel and simple compositional model for abstracting complex dynamics using a hierarchy of transition functions;

2. A new algorithm to learn hypernetworks that can generate transition functions on the fly using prediction errors;

3. A scalable, compositional and interpretable world model geared towards efficient use of abstract labels for faster planning and transfer in real-world environments.

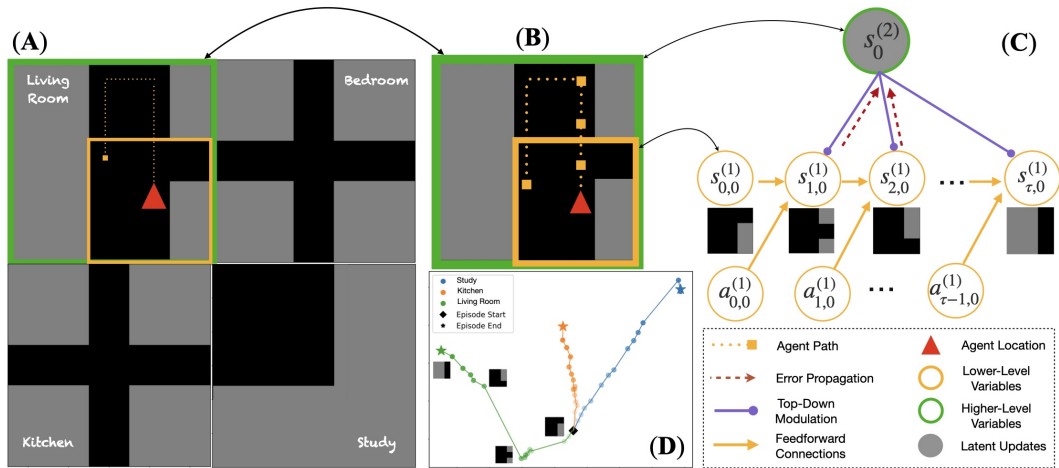

Figure 2: **Inferring higher-level abstract states from lower-level dynamics**. **(A)** A home environment composed of simpler and, possibly repeating elements ("rooms"). Gray areas represent walls or regions unreachable by the agent. The rooms are separated to show independent dynamics and an opportunity for reuse of transition functions between similar rooms (e.g., parts of the bedroom and kitchen). **(B)** The rooms correspond to reusable local dynamics that can be abstracted as the higher-level latent state vector $s^{(2)}$ and inferred by Composer. **(C)** Latent state inference by unrolling the state transition graphical model over time and integrating temporal information. **(D)** 2-D TSNE plot of successive updates to a $d = 32$ dimensional latent state vector, while the agent explores a room for $\tau = 15$ time steps. Note that the inference process converges to different parts of the latent space for different rooms. More examples in Supplementary Section.

## 2 LEARNING AND INFERRING ABSTRACT TRANSITION DYNAMICS

In this section, we demonstrate how state abstractions and hierarchical transition functions can be learned by considering a 2-level Composer model (Figure 1; Figure 3). We leverage variants of hypernetworks which are neural networks that generate the parameters of other neural networks (Ha et al., 2016) to carefully study different properties of top-down modulation and dynamics abstraction in simple compositional gridworld environments. Figure 3 shows the parameterization of hypernetwork based Composer model. Mention gridworld being top down POMDP.     FIX

### 2.1 TOP-DOWN MODULATION

For a simple implementation of hierarchical abstraction of dynamics, we consider two possible approaches. Both approaches use an approximation of hypernetworks. In our first approach, a hypernetwork predicts a vector with $K$ weights $\mathbf{w} = [w_1, w_2, ..., w_k]$ for combining a set of learnable basis matrices $\mathbf{M}$, generating the state transition function $f_s$ at the lower-level. This approach is similar to prior work in abstracting temporal neural signals (Jiang & Rao (2022a)).

$$\mathbf{w} = \mathcal{H}(\mathbf{s}_T^{(2)}) \tag{1}$$

$$f_s = \sum_{k=1}^{K} w_k \mathbf{M_k} \tag{2}$$

$$\hat{s}_{t+1} = ReLU(f_s(s_t, a_t)) \tag{3}$$

We also experiment with an embedding approach for top-down modulation where the hypernetwork predicts a vector embedding from the higher-level latent state. The set of matrices $M$ is replaced by an RNN that takes as input the top-down embedding, the current state and action as inputs and predicts the next state. In practice, we found that adding additional decoders after the RNN prediction in this approach, gave results comparable to the mixture of matrices method discussed above.

$$\mathbf{e} = \mathcal{H}_\theta(\mathbf{s}_T^{(2)}) \tag{4}$$

$$\mathbf{h_t} = \tanh(\mathbf{x_t}W_1 + b_1 + \mathbf{h_{t-1}}W_2 + b_2) \tag{5}$$

$$\hat{s}_{t+1} = ReLU(\mathbf{h_t}W_3 + b_3) \tag{6}$$

Where $\mathbf{x_t} = [\mathbf{e}, s_t, a_t]$ is the concatenated input at the lower-level and $[\theta, W_{1:3}, b_{1:3}]$ are the model parameters. Unlike traditional autoencoders (Kingma & Welling (2013); Baldi (2012)), this model does not have an explicit encoder mapping observations to a latent space. The abstract vectors are directly inferred via backpropagation of prediction errors during inference (rather than being used solely for learning as in traditional neural networks).

## 2.2 INFERENCE

Consider an agent exploring its environment using actions defined by an exploration policy $\pi$. To make the example more concrete, assume that the agent is in a home environment made of rooms (kitchen, bedroom, etc.), as shown in Figure 2(A). A sequence of observations are generated from the sensory apparatus of the agent as it explores the environment. We assume that the underlying states are partially observable, resulting in a trajectory of observed states and actions over $\tau$ timesteps: $\mathcal{T}_{a \sim \pi} = \{s_0, a_0, s_1, a_1, ..., s_\tau\}$ [1]. Throughout the paper, we assume that the internal states $s_t$ are based on encoded representations of inputs $x_t$ (Figure 1) and integrate historically observed inputs via the recurrent network, a formulation in line with the recent trends in model-based RL (Hafner et al. (2019; 2020)). However, in our model, this recurrent network (which implements the lower-level transition function) is generated on the fly by the current higher-level abstract state $s^{(2)}$. Formally, $s_{t+1} \sim P(s_{t+1}|s_t, a_t, s^{(2)})$. **Notably, even $s^{(2)}$ is unknown and must be learnt directly from the environment**. We pose the process of learning abstraction vectors $s^{(2)}$ as a continuous inference process in time, similar to estimation in kalman filters. Since our hierarchical transition models are task-independent, the rewards obtained in any particular task do not directly affect the transition models. We intend to explore incorporating reward prediction (in addition to state prediction) at the lower level in future work (Hafner et al. (2020)).

The inference process involves making updates to beliefs over the higher-level state $s^{(2)}$ (e.g., what room the agent is located in, i.e.. a kitchen, bedroom, etc.) as evidence accumulates over an episode. This corresponds to inference of $s^{(2)}$ by minimizing prediction loss using gradient updates for each lower-level time step $t$ (Equations 7, 8; Figure 2(C)):

$$\mathcal{L}_{t,s^{(2)}} = ||\hat{s}_{t+1}^{(1)} - s_{t+1}^{(1)}||_2^2 + \lambda||s_T^{(2)}||_2^2 \tag{7}$$

$$s^{(2)} \leftarrow s^{(2)} - \eta \nabla_{s^{(2)}} \mathcal{L}_{t,s^{(2)}} \tag{8}$$

The first term in equation 7 is the prediction loss. We typically use a decoder to transform the recurrent network predictions to the original observation space. The second term is an $L_2$ regularizer on the abstract states which we found improves performance. $\eta = 0.05$ is the inference learning rate which is kept higher than the model parameter learning rates. During the above inference process, no update is made to the model parameters (Figure 3(A)). A TSNE plot of $s^{(2)}$ converging over specific episodes to represent different rooms is illustrated in 2(D). We investigate properties of the latent $s^{(2)}$ space in Section 4.

## 2.3 LEARNING

Learning the parameters of the hierarchical model is straightforward (Figure 3(B)). After running the inference process for $\tau$ steps, latent states $s^{(2)}$ are frozen and used as inputs to the hypernetwork. For the same set of observations used during inference, prediction errors for the $\tau$ timesteps are accumulated and the model parameters are updated in an unsupervised manner.

---

[1]For convenience and readability of equations, we omit the subscripts and superscripts for variables throughout the paper, unless necessary: $s_{t,T}^{(1)}$, the lower-level state at time $t$ and at a higher-level time period $T$, is replaced with $s_t$. Similarly, the higher-level state $s_T^{(2)}$ is replaced with $s^{(2)}$, when $T$ remains constant.

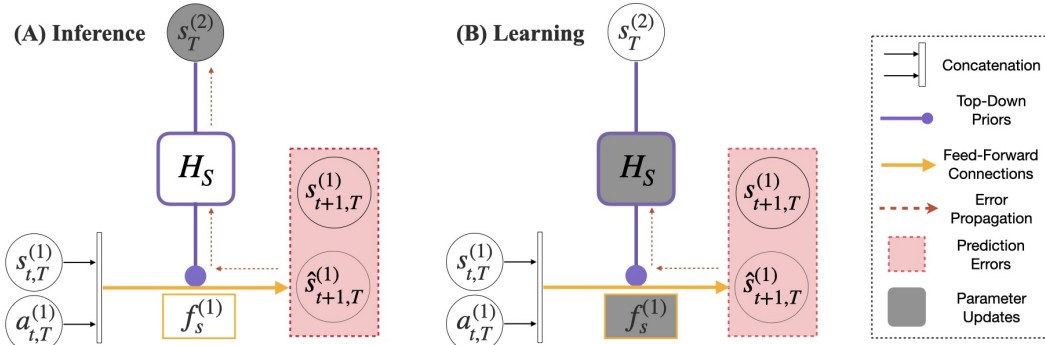

Figure 3: Top-down modulation with Hypernetworks. **(A)** During inference, gradient updates to all the model parameters are switched off, except the higher-level latent code. Next state prediction errors accumulate and modify the latent via the backpropagation algorithm. **(B)** After running $K$ inference steps, the latents are frozen and the model parameters are updated. The inputs to the transition function are the current state and action of the agent. Note that we do not fix a target $s^{(2)}$ vector. It is learnt along with the model parameters in an alternating gradient descent manner.

$$\mathcal{L}_\theta = \sum_{t=1}^{\tau} ||\hat{s}_{t+1}^{(1)} - s_{t+1}^{(1)}||_2^2 \tag{9}$$

$$\theta_\mathcal{H} \leftarrow \theta_\mathcal{H} - \eta_\mathcal{H} \nabla_{\theta_\mathcal{H}} \mathcal{L}_\theta; \ \ \theta_f \leftarrow \theta_f - \eta_f \nabla_{\theta_f} \mathcal{L}_\theta \tag{10}$$

## 3 SCALING COMPOSER WITH CONTRASTIVE LEARNING

The real test for Composer is when it is scaled to realistic, pixel based, ego-centric environments. We choose Habitat 2.0 for their fast and efficient rendering, flexibility in environment configuration and their highly realistic suite of embodied AI tasks (Szot et al., 2021). Habitat's noisy and sometimes partially occluded images provide a very realistic scenario for benchmarking our approach.

### 3.1 WHAT DOES COMPOSER LEARN IN HABITAT 2.0?

There are two fundamental difficulties when scaling our originally proposed algorithm seen in Section 2. First, inferring abstractions from prediction errors, and then subsequently training the model in a alternating descent fashion is very inefficient and slow. The entire process requires a minimum of two backpropagation step and often more if we choose to infer for multiple steps. Modern auto-differentiation libraries like pytorch (Paszke et al., 2019) and parallelized GPU operations render the proposed Composer algorithm much slower than the state-of-the art hierarchical RL approaches.

Second, and more importantly, early experiments show that it is not trivial to learn well defined and well separated abstractions in complex, and noisy RGBD images. As seen in Figure 4, with increase in complexity of the sequential data, the ability to discern different abstract dynamics diminishes. In Figure 4(C), the agent was allowed to explore the entire environment without supervision. We see that even though there are regions of densely clustered points for a given room, it is hard to decipher anything meaningful and reliable when several complex dynamics are involved. This problem is compounded by the fact that it is not obviously clear if there is an inherent difficulty in learning dynamics for this environment or if the model is utterly distracted by spurious variations, a feature of visual pixel based reconstruction (Stone et al., 2021; Zhu et al., 2023).

To address the first issue, we introduce an encoder RNN to directly estimate abstract states $s^{(2)}$ from lower-level variables. We can now avoid an extra inference step which required propagating error gradients across the model. To address the second issue, we introduce supervised contrastive learning in the next subsection.

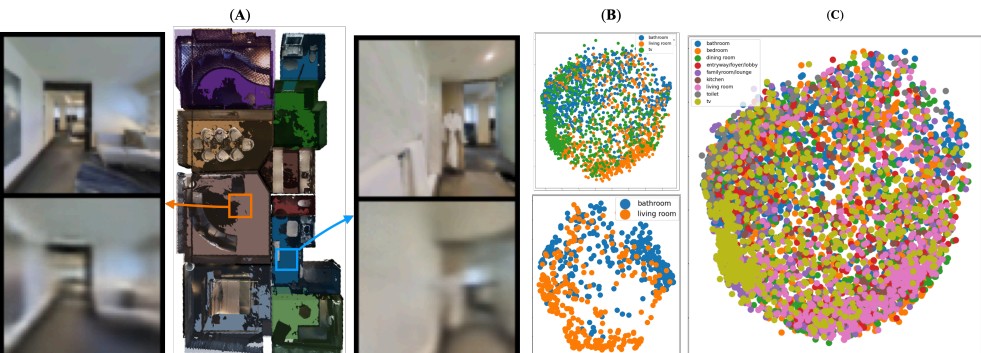

Figure 4: **Abstract state inference on Habitat**: Dynamics abstraction and next state prediction with $s^{(2)}$ as prior. The scaled APC model (Figure S10) was trained on randomly generated trajectories from Habitat 2.0's ego-centric home environment. **(A)** Rendering of a home used in our experiments. The render was taken from Matterport (Chang et al. (2017)). The next step predicted model outputs (bottom) and ground truth reconstruction targets (top) are shown for two different rooms. **(B)** 2D PCA of inferred $s^{(2)}$ vectors after training Composer for episodes starting at 2 and 3 different rooms. With no training signal apart from dynamics prediction errors, $s^{(2)}$ shows moderately separable clusters for different rooms. **(C)** 2D PCA of inferred $s^{(2)}$ vectors after training the model for all rooms in the environment.

Supplementary Figure S10 depicts a version of the Composer model which relies on an encoder RNN that directly infers an abstract state and modulates another lower-level dynamics prediction via the network $H_s$. Prior work (Galanti & Wolf (2020)) has shown that such an embedding input-based approach is functionally equivalent to using a hypernetwork. Additionally, instead of inferring $s^{(2)}$ via backpropagation of prediction errors, this version of the model uses a simple feedforward encoder to directly infer $s^{(2)}$ from a sequence of image inputs (amortized inference), leading to significant improvements in training time and parallel processing.

## 3.2 SUPERVISED CONTRASTIVE LEARNING

To overcome the non-trivial problem of learning well defined and well separable abstractions, we turn towards the human brain. We as humans, do not necessarily learn every abstraction bottom-up from scratch. More often than not, we strongly rely on signals and labels used by others and quickly adopt them to our own internal models. This is especially true when exploring a novel scene or a problem. Humans try their best to use existing ideas and concepts to derive new solutions (Lake et al., 2016).

Motivated by this fact, we look towards learning from labels. Recent advances in Vision Language Models, have made it incredible cheap to generate and gather labeled images and videos (Radford et al., 2021; Deitke et al., 2024; Liu et al., 2024). Supervised Contrastive Learning is a variant of SimCLR that uses labels to learn robust and powerful representations of visual data (Khosla et al., 2021; Chen et al., 2020). Given some anchor indices $I$, if $P(i)$ are the set of positive examples for anchor $i$ (samples from same room in Habitat, for example) and $A(i)$ is all samples excluding the $i^{th}$, the supervised contrastive loss for a batch of inputs is defined as:

$$\mathcal{L}_{\text{sup}} = \sum_{i \in I} \frac{-1}{|P(i)|} \sum_{p \in P(i)} \log \frac{\exp(\mathbf{z}_i \cdot \mathbf{z}_p / \tau)}{\sum_{a \in A(i)} \exp(\mathbf{z}_i \cdot \mathbf{z}_a / \tau)} \tag{11}$$

Here, $z_i$ is the representation of the $i^{th}$ example and $\tau$ is a scalar temperature parameter. We incorporate this loss along with reconstructions for our world model. For Habitat 2.0, we use the room labels provided by the environment. Deriving labels from a state of the art VLM is another possible approach. We show that using an extremely small number of labels (less than 2% of training steps) is sufficient for inferring well separable dynamics and further modulating a world model.

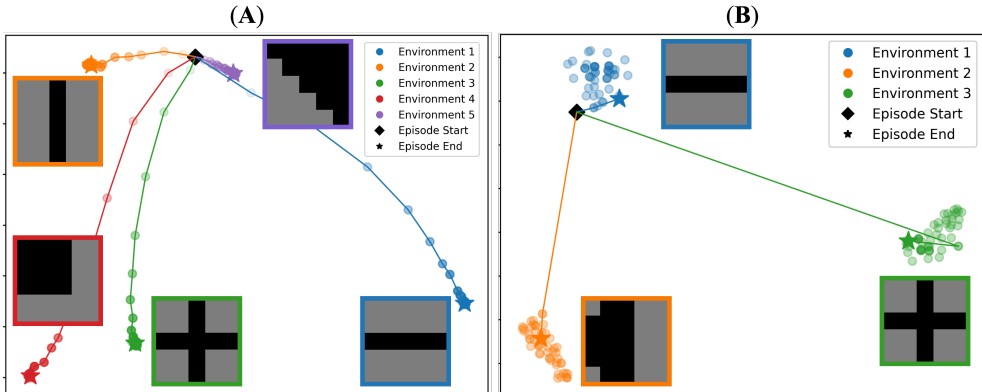

Figure 5: Inferring the 5 x 5 room with partially observable patches. Both the figures are 2-D PCA of $d = 32$ dimension latent codes. Note that the model-based agent does not know which room it is put into, and must infer from observations. **(A)** $\tau = 10$ step inference for different environments trained with episodes of length 10. **(B) One-Shot inference** for three different rooms. Final latent codes for 100 random episodes from each environment are also plotted in the background to validate the one-shot inference. Details in text.

## 4 EXPERIMENTS AND RESULTS

### 4.1 ABSTRACT TRANSITION SPACE

Figure 5(A) shows the inference process for different gridworld rooms after training the Composer dynamics model. Random trajectories of length $\tau = 15$ are drawn from different rooms and used for dynamics prediction task. Accurate estimates can be made in time, as the agent gathers more evidence. Figure 5(B) shows one-shot inference for three rooms when the model is trained with shorter episodes of length $\tau = 5$. In the same figure, we also plot the PCA of final abstract states from 100 episodes for each environment. This shows that the one-shot inference result is indeed accurate. This fast inference method is useful for rapid planning with limited data. Details are in the Supplementary Section A.3.

### 4.2 ZERO-SHOT TRANSFER TO NEW ENVIRONMENTS

A significant benefit of abstracting transition dynamics into a continuous latent space is fast transfer to new environments. To illustrate this, we trained the hierarchical dynamics model on two simple environments - a vertical and a horizontal hallway. Figure 6 shows the PCA of the higher-level abstract state space with blue and orange clusters representing the final inferred abstract states for the environments. We sampled points along the line joining the cluster centers and used the points as priors to generate a transition function at the lower level. Next state predictions were made using the generated function and a random policy. These predictions were used to reconstruct the dynamics and hence the environment captured by the transition function. These new environments ("rooms") are plotted as $5 \times 5$ grids in Figure 6. These new rooms, which were never seen by the model, demonstrates how the model can compose and transfer learned dynamics to new environments.

### 4.3 STATE ABSTRACTION IN HABITAT 2.0

We now show that Composer can be scaled to learn abstract states on realistic environments using very sparse labeled data and supervised contrastive learning. We use a modified Composer as seen in Figure S10). Dealing with high-dimensional image inputs require powerful encoders and decoders. For this experiment, we use a pretrained Residual Autoencoder, resnet18 (He et al. (2015); Wijmans et al. (2020)) to encode and decode $256 \times 256$ RGB and depth images from Habitat 2.0 Savva et al. (2019); Szot et al. (2021) (replacing autoencoders with transformer-based ViT (Vaswani et al. (2017); Dosovitskiy et al. (2021)) or other architectures is straightforward). We task the scaled model to predict single-step future states, given the current inputs and actions, and train for 500

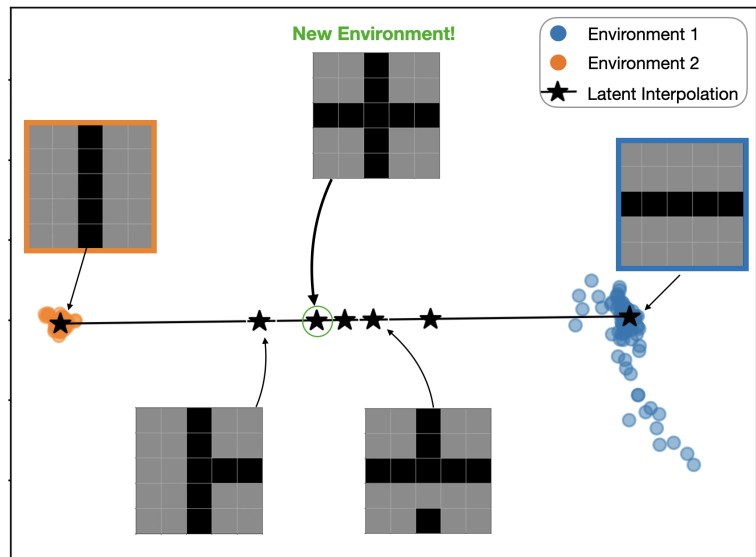

Figure 6: **Zero shot transfer of dynamics** to new environments by interpolating the abstract states $s^{(2)}$. The dynamics for newly sampled abstract states (priors) are inferred from a model trained only on Environment 1 and 2 (Note that these rooms have different dynamics in a top-down setting). The inferred dynamics with interpolated $s^{(2)}$ priors are drawn out as new environments. This hints that the priors could be learning a smooth space spanning continuously changing transition functions.

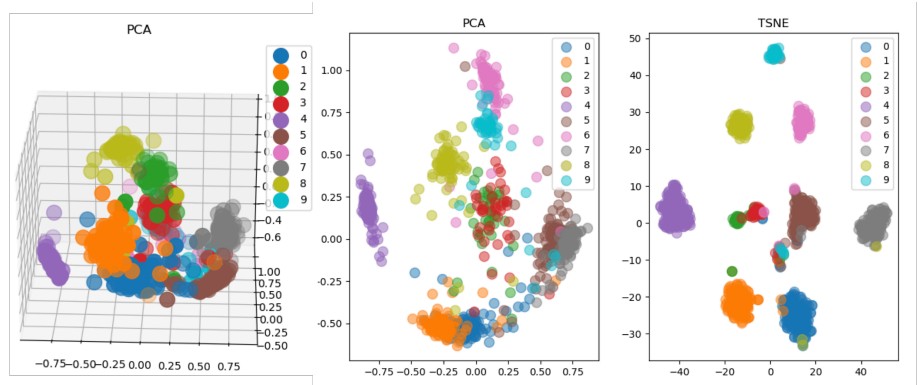

Figure 7: **Abstractions learnt via Contrastive Learning**: Composer learns well defined and human interpretable abstractions. These abstract states are used to modulate a lower level transition dynamics. Reconstruction figures in the appendix. The abstract vectors were of size 32 dimensions. The first 4 principal components explain 80% of the variance. More importantly, only 2% of the samples were labeled with information about the agent's room.

epochs. The results without contrastive learning are shown in Figure 4 and with contrastive learning are shown in Figure 7 and Supplementary Figure S11.

## 4.4 LEARNING HIGHER LEVEL TRANSITION MODEL AND ACTION ABSTRACTIONS

As discussed above, the higher-level state $s^{(2)}$ abstracts the transition dynamics at the lower-level (using backpropagation of prediction errors or an encoder). Significant efficiencies can be achieved by learning a transition function between abstract states, allowing higher-level planning and navigation to any goal in a compositional environment like Figure 2(A). To learn a transition function between abstract states, we introduce the idea of an *abstract action* $\mathbf{a}_T^{(2)}$ (similar to an "option" in hierarchical RL) which is a latent action vector that generates a lower-level policy. We can de-

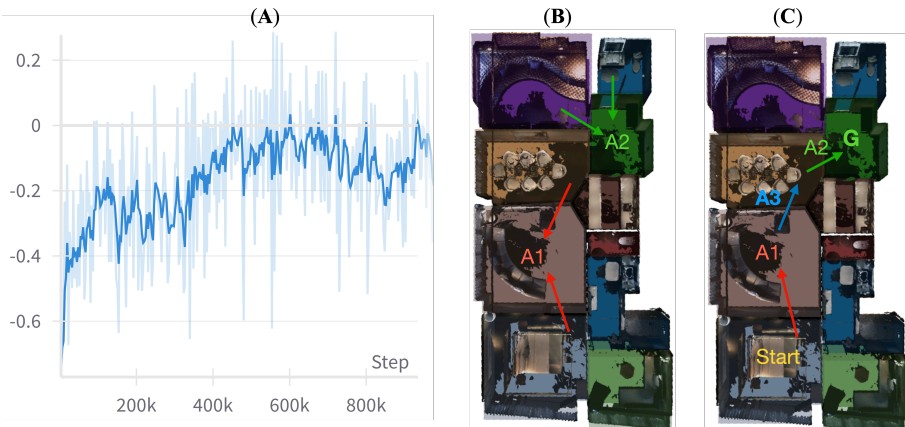

Figure 8: **Rooms are subgoals**: We consider the learnt abstractions $s^{(2)}$ as subgoals in themselves. Policies to reach these subgoals can be learnt and composed to reach any novel goals in the room. **(A)** Reward learning curve for a goal conditioned DQN agent. The rewards are internal to Composer model and the optimal policies are learnt after a few warm-up epochs of world model learning. **(B)** Learnt abstractions can be used as subgoals. In case of habitat, the abstractions are rooms. A1 in the above figure is the living room and A2 is the longue. Policies that reach these rooms are learnt in a sample-efficent manner with the same data used by the world model. **(C)** The learnt policies can be chained to reach a goal (G) previously unseen by the agent.

fine action abstractions in Composer to represent subgoals or subtasks similar to the formulation in (Hafner et al. (2022); Schmidhuber (1991a); Abel (2022)). These abstract actions are tied to a context dynamics, since a particular action might not be relevant in all scenarios. For example, "Open the microwave" is a valid subgoal if the agent context is kitchen and not when the context is, say, a conference room. The latent codes for action abstractions can be learnt by a similar inference process discussed for state abstractions.

For this paper however, we consider the state abstractions to also represent subgoals. We leave the subgoal learning with Composer as a future work and instead learn a policy, conditioned on $s^{(2)}$ as subgoals represented by one hot vectors. Figure 8 shows these learnt policies for subgoals of Habitat environment. Given abstract actions, the transition dynamics for higher-level abstract states can be defined as $P(\mathbf{s}_{T+1}^{(2)} | \mathbf{s}_T^{(2)}, \mathbf{a}_T^{(2)})$, implemented by a recurrent network $f_s^{(2)}$, where $T$ represents a time step at the higher level in the hierarchy.

## 4.5 HIERARCHICAL RL AND PLANNING

It is well-known that the learnt abstract states, along with well-defined abstract actions, can reduce the effective search space of an agent for reinforcement learning and planning, greatly reducing the complexity of the problem (Nachum et al. (2019)). To demonstrate that this advantage accrues to Composer, we performed simple experiments on a compositional gridworld environment (Figure 2). A simple instantiation of Composer's hierarchical transition model and hierarchical policy was learnt for this environment. Here, the abstract actions were assumed to be one of 8 possible subgoals. We considered 2 tasks (1) Goal-reaching RL task where the goals can change at any point in time, and (2) Planning to reach a fixed goal from increasing distances. The baselines for these experiments are respectively: (1) A policy gradient model-free agent and (2) An MPC planner with full access to the oracle transition dynamics. Our results (Figure 9) show that Composer is indeed robust to goal changes and can plan faster, as long as the abstract actions are well defined. Work on learning useful skills without hand-designed abstract actions (Eysenbach et al. (2018)) is ongoing.

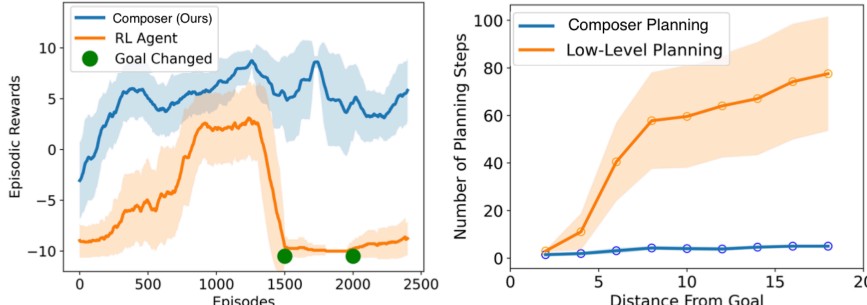

Figure 9: Planning with Composer on the grid world environment. (**Left**) Rewards collected over episodes as the goals are changed. RL agents are not robust to changing goals. (**Right**) Action steps taken to plan. With abstract actions, Our model can plan exponentially faster due to the reduced sequence length.

## 5 CONCLUSION

This paper presents a new method called Composer for learning transition dynamics for complex real-world environments based on a structured heirarchical model. The method is inspired by the theory of the mammalian cortex, and learns a hierarchy of transition functions using self-supervised learning based on prediction errors and hypernetworks. We applied the model to both traditional grid worlds and the more complex Habitat domain and showed that higher-level latent codes that generate transition dynamics for different environments form different clusters in the latent space. Furthermore, this continuous latent space exhibits smooth transformations of transition functions, allowing Composer to generate dynamics for new environments in a compositional manner. We introduce abstract actions to allow transition functions to be learned for higher-level latent state spaces, giving rise to hierarchical world models. We also introduce contrastive learning with very sparse labels to regularize the learnt abstractions and align them towards human interpretable representations. Our results demonstrate the efficacy of higher-level planning using Composer by exploiting learned hierarchical world models and local reference frames. Our ongoing and future work is focused on scaling Composer to larger-scale environments and RL benchmarks, and leveraging the model's compositional structure and ability to generate new transition functions on the fly to achieve fast transfer across environments.

## 6 REPRODUCIBILITY STATEMENT

Throughout the paper, model details and relevant equations are discussed in detail. Numerous diagrams and plots are shown to clearly explain the core insights behind the paper. Section 2 mentions all equations required to reproduce the results. We provide precise experiment setup for abstraction inference in Supplementary Section A.3. We commit to releasing anonymous source code during rebuttal phase. The current state of the Composer codebase is not in line with the double blind policy. All our code was run on a single RTX 4090 24 GB GPU and hence is highly reproducible even by the most modest computing resource.

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

# A APPENDIX

## A.1 RELATED WORK

The most closely related work is the Active Predictive Coding model (Rao et al. (2023)) and related papers on predictive coding Rao & Ballard (1999); Jiang & Rao (2022a). Other related work pertains to various components and aspects of our model including: (1) Top-down Modulation, (2) Hierarchical World Models (3) Hierarchical Policy (4) Reference Frames.

**Top-Down Modulation** refers to an abstract state, conditioning a function that is usually working at a constrained spatio-temporal scale. Intuitively, the latent vector abstracts critical parts of the lower-level transition or policy function, allowing re-use of learned dynamics or policies in novel scenarios. For example, a child learns to lift a coffee mug and has no problem transferring that experience to picking up a jar or pitcher with a handle. There is evidence from neuroscience that the cortex may use top-down gain modulation to facilitate such transfer of learned behaviors (Ferguson & Cardin (2020)). We propose variants of hypernetworks (Ha et al. (2016); Galanti & Wolf (2020)) as potential candidates to implement such abstractions in the Composer model.

**Hierarchical Transition Models**: Ha & Schmidhuber (2018) introduced world models into model-based RL. Since then, powerful variants of world models have been proposed for modeling increasingly complex environment dynamics (Hafner et al. (2020; 2022); Micheli et al. (2023)). Yet, these world models are limited in scope when exposed to novel environments. Graph schemas have gained in popularity in recent years as potential computational mechanisms for emulating abstraction, transfer and planning in the brain (Guntupalli et al. (2023); Moser et al. (2017); Whittington et al. (2021)). Our model, which is inspired by the brain's hierarchical architecture, employs hierarchical world models that learn abstractions of transition functions limited in space and time, and further learns to transition in the new abstract space with access to only unsupervised prediction errors.

**Hierarchical Policies**: Hierarchical Reinforcement Leaning and action abstractions have have a long history in RL (Sutton et al. (1999), Barto & Mahadevan (2003), Schmidhuber (1990; 1991a;b)). With the introduction of deep neural networks (LeCun et al. (2015); Schmidhuber (2014)), many variants of hierarchical and deep reinforcement learning have been developed (Bacon et al. (2017); Hafner et al. (2022); Kulkarni et al. (2016)). Abel (2022) provides an extensive discussion of abstract states and actions. In Composer, a higher-level abstract action vector is generated by the higher-level policy, and this action vector in turn generates, via a hypernetwork, a low-level policy function; details in Section 3.4, see also (Rao et al. (2023))).

**Reference Frames**: Our approach decomposes a complex problem into transition functions and policies that operate hierarchically within local reference frames. This allows an agent to ignore task-irrelevant state and action information at each level, resulting in considerable efficiencies in training and transfer. The importance of reference frames in intelligence and reasoning has been highlighted recently by Hawkins (2021) based on evidence from neuroscience involving "grid cells" and spatial reference frames in the cortex and hippocampus (O'Keefe & Dostrovsky (1971); Moser et al. (2017)). Previous work in AI on hard attention models (Mnih et al. (2014)) can be regarded as single-level versions of our approach which learns hierarchical reference frames (Section 2).

## A.2 EXPERIMENT AND MODEL DETAILS FOR THE SCALED COMPOSER

See Supplementary Figures S10 and S11

## A.3 ADDITIONAL RESULTS: FEW-SHOT INFERENCE OF ABSTRACT STATES

Here, we provide experiment details for the analysis of abstract state inference. Results are shown in Figure 5. Our experiments with the learnt abstract states $s^{(2)}$ was focused on studying the nature of the abstract transition space. This turned out to be useful when transfering learnt dynamics to novel environments. For our experiment setup, we collect episodic data for 5 room environments with different dynamics. The environment dynamics can be changed by placing the walls in different patterns. The hypernetwork used is a 4 layer deep neural net with 256 units at each layer. We use ReLU non-linear activation everywhere unless specified. The learning rate for inference was

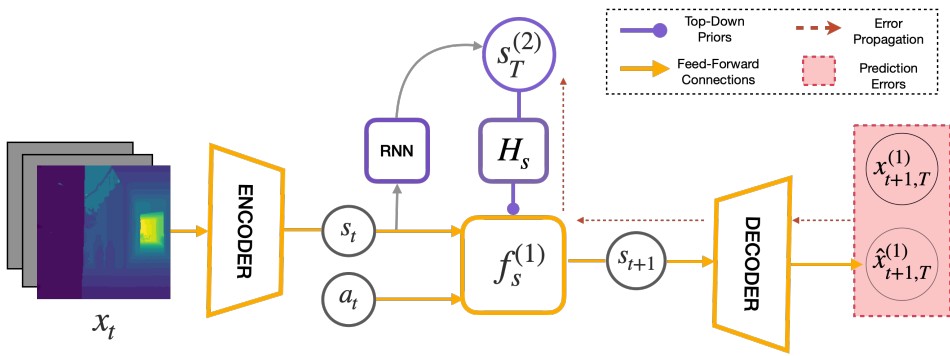

Figure S10: **Scaling Composer to complex image-based environments**. The core idea of using abstract states $s^{(2)}$ to generate lower-level transition dynamics $f_s$ remains the same but instead of relying on prediction errors for inference, a bottom up encoder is used for amortized inference to directly infer the abstract state from the accumulated lower level evidence. Pixel based depth images of size (256 x 256 x 1) from habitat are fed into Composer and $s^{(2)}$ estimates are computed at each time step. Additionally, we regularize $s^{(2)}$ with sparse labels and supervised contrastive learning.

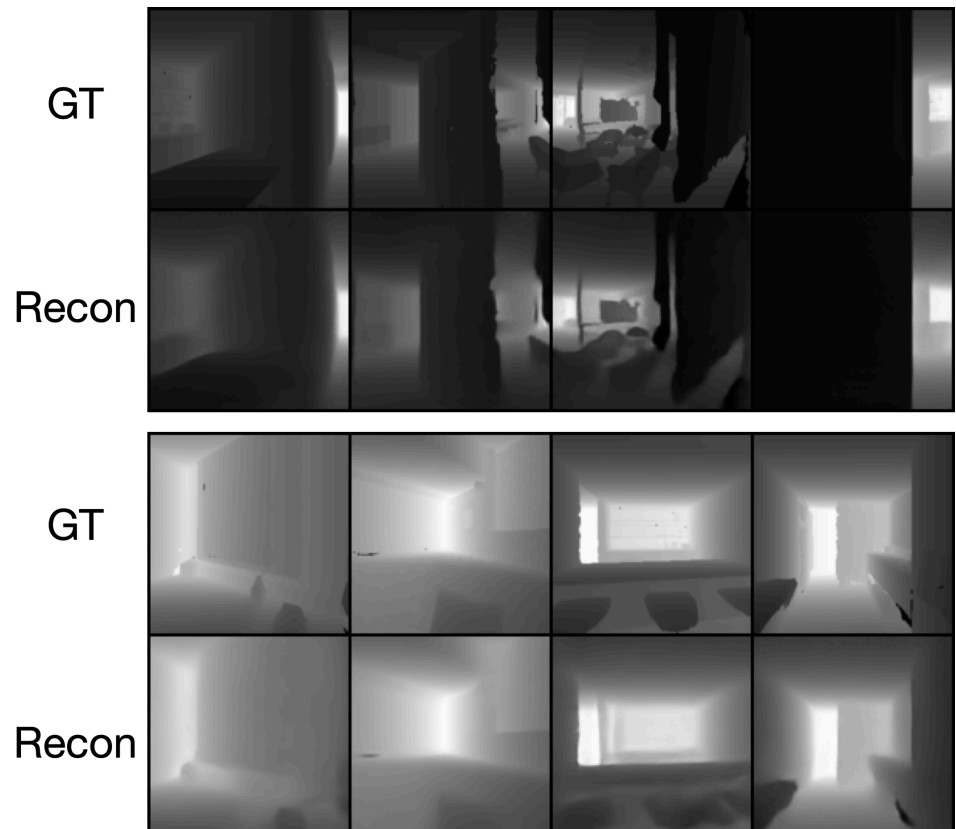

Figure S11: **Composer depth reconstructions for an agent**: Along with highly accurate reconstructions, the model is also able to infer abstract states like kitchen, room, etc.

kept much higher at $\eta = 0.1$, whereas the learning rate for training both the hypernetwork and the transition function were $\eta_{\mathcal{H}} = \eta_f = 0.001$. For each environment, we collect episodes of length $\tau$ and feed it to Composer model. We experiment inference with episodes of lengths 2, 5, 15, 25 and 50. Typically, longer episodes perform better since the data available about the environment increases. For every episode, our model first infers the latent code, freezes the final latent code and performs gradient updates to the model parameters using the prediction errors. Adam optimizer was used for both inference and training.

To choose a dimension for $s_{(2)} \in \mathcal{R}^d$, we run inference and training for $d = [4, 8, 16, 32, 64]$. Our intent was to create a balance between information capacity (neatly clustered latent codes) and prediction errors. ( Dawid & LeCun (2023)) notes that generative latent variable architectures can collapse if the latent codes have very high information capacity. In such cases, the transition function completely ignore the inputs $s_t, a_t$ and learn to essentially push all the necessary information into the latent code. In our experiments, we found $d = 32$ to optimally minimize prediction errors while maintaining separable latent code clusters. The plots representing latent codes in this paper are 2-D PCA of $s^{(2)}$ originally in a 32 dimension space unless specified.

