# OpenReview forum: "Compositional World Models with Interpretable Abstractions"
_ICLR.cc/2025/Conference — ICLR 2025 Conference Withdrawn Submission_

### Official Review · Reviewer_JL4b · 2024-10-23

**Soundness:** 2
**Presentation:** 1
**Contribution:** 1
**Rating:** 3
**Confidence:** 4

**Summary:**

The authors present Composer, a method for learning a hierarchical transition function where the low-level prediction of the next state depends on a high-level latent variable. The model uses hypernetworks, some self-supervised learning and some supervised learning from labels and is applied to a toy gridworld and to some Habitat 2.0 scenes.

My decision is to reject the paper as it is far from ready for publication.

**Strengths:**

- the general question addressed by the authors of learning useful state abstractions to get a hierarchical transition function is very relevant

**Weaknesses:**

- the paper is unclear about its objectives: some studies performed in Sections 4.4 and 4.5 do not correspond to the objectives mentioned in the introduction, and the necessary elements to perform these studies are not described in the methods. The figure corresponding to what should be interpreted as the main contribution is rejected into an appendix. I suggest moving Fig. S10 back to the main paper and reconsider the introduction so that it incorporates the objectives of Section 4.4 and 4.5 (or refocus the paper and remove these sections).
- it is not clear whether the paper has a body of results that could be the contribution of a good paper, given that most of the experiments and conclusion build more on ongoing work and things to come and most of the results appear mainly preliminary. The authors should focus more on strengthening and expanding the completed results, providing more in-depth analysis and discussion of their significance.
- the methods are insufficiently described and not clearly formalized. See below for details and advices.
- the paper does not have a related work section. The authors should include a comprehensive related work section that covers key areas such as hierarchical reinforcement learning, world models, and neuroscience-inspired AI approaches. This will help contextualize their contribution within the field.
- the Composer system is not compared to any baseline. The authors should compare to relevant work in the experimental section.
- there is an ablation of removing the supervised contrastive loss, but this is the only ablation and it is not identified as such.

**Questions:**

I’m using this section more to criticize the current form and to suggest improvements to the authors, as I think the paper is too far from being ready for publication.
- the authors should ask themselves who did things similar to what they are trying to do, then write a related work section and compare their approach to baselines. This is only through such comparisons that we can determine whether their work is a useful piece of research or not. The answer “we are the only ones to tackle this question is always wrong. For instance, the authors should have a look at this paper:
Gumbsch, C., Butz, M. V., & Martius, G. (2021). Sparsely changing latent states for prediction and planning in partially observable domains. Advances in Neural Information Processing Systems, 34, 17518-17531.
and other papers from the same authors. I’m quite sure that they will find many works they should compare themselves to.
- discovering the high level latent state is obviously the difficult question in the author’s setup. One expects some clever new idea to do this when reading the abstract and the introduction of the paper. But this is only in page 5 that the authors mention for the first time that they will use supervised contrastive learning. This appears as a late and unsatisfactory addition to their model after (probably) failing to use anything requiring less human engineered data. The authors should definitely be honest about their method since the beginning, as they generate expectations that are not fulfilled. The argument “but humans also learn from labels” to counteract the negative impression it generates is also very weak and makes the situation even worse.
- the formalization is far from satisfactory.
* footnote 1 p. 4 specifies that there is a high level time period T which is not introduced and never described. How does the high level time change. It is defined manually? The authors should start section 2 with a problem statement where they describe the setup and all their assumptions
* the equations should come inside a sentence explaining what they are about
* In (1) H is the hypernetwork, right? This should be mentioned line 148. In (4) it is noted H_\theta…
* Eq. (4) describes e, but e is not used anywhere anymore
* We have to guess that the authors will use Eqs (4) to (6) rather than [1) to (3), this is not clearly stated
* Line 182, the authors mention using a recurrent network that has never been described (nor any hyper-parameter of the method, by the way). This is where we guess that they use (4) to (6)
* In (8), we do not know what s_T is, footnote 1 does not help much. Is the lambda term a regularizer? This is not explicit at all.
* Make a sentence to describe what (9) and (10) are about.
- lines 186 sq. : “Since our hierarchical transition models are task-independent, the rewards obtained in any particular task do not directly affect the transition models. We intend to explore incorporating reward prediction (in addition to state prediction) at the lower level in future work (Hafner et al. (2020)).” → this should move to a future work section (as many other statements about ongoing or future work…)
- Figure 4, the caption should conclude about what we should see from the right part. Actually, I would put the version with the contrastive loss first, and the ablation later in the paper.
- In Figs 5 and 6, does the x,y position of image patches mean something, or is it only their relative distance that matters? Would we get the same organization if we had many more patches, as should be the case for Habitat 2.0?
- line 376: “replacing autoencoders … with ViT … is straightforward”: so why didn’t the authors do it?
- In Section 4.4, the authors introduce abstract actions, subgoals, higher-level transition dynamics, but the description is rather incomplete. Shouldn’t these elements be presented in the methods? Or is it just a side result? If it is a side result, shouldn’t it be published in a side paper?
- In Section 4.4 the authors state that subgoal learning is left for future work, but in Section 4.5 there are 8 possible subgoals, we do not know where they come from. Again, the authors should have a clear problem statement in the beginning of Section 2 to delineate the problem they want to address and their assumptions, and then stick to the problem they have described.
- line 484 “Work on learning useful skills without hand-designed abstract actions (Eysenbach et al. (2018)) is ongoing.” Such a sentence should not appear in a results section. Maybe in future work, but the best is to get the corresponding results, then publish them.
- Figure 9(a), why are the episodic rewards decreasing BEFORE the goal changed? This needs to be commented upon.
- line 506: “The method is inspired by the theory of the mammalian cortex”. If there was such a unique theory, I would be glad to know it. The authors probably mean “the predictive processing theory...”, but they have to be aware that this is not the only theory. Furthermore, in the introduction where the authors shortly describe some elements of this “theory”, they call upon various corresponding to various perspectives, I’m not sure we can consider the corresponding set of elements to constitute a theory. And again, the authors should compare themselves against other models that are inspired by these various elements of this “theory”.
- Figure S10 should be moved into the main paper, if the main results are about Habitat 2.0

# Typos, minor errors:

- the authors mention many times that their method uses unsupervised learning, but it seems more “self-supervised”, at it self corrects its prediction based on the posterior evidence.
- why not call your latent variables “z”, as many authors do?
- line 96: code snippets are promised in the Appendix, but I could not find them
-line 142: “Mention gridworld being top down POMDP. FIX” A good sign that the paper is not ready for publication…
- refer to equations using "eqref{}" rather than "ref{}"
- equations finishing a sentence should finish with a dot.
- line 182: a formulation in line with… : what do the authors mean: that it vaguely ressembles...? the authors have to be more accurate.
- kalman → Kalman
- line 255 two … step(s)
- line 298 (amortized inference): make a sentence. What do you want to say?
- line 367 These new rooms … demonstrates. Apart from the grammar issue, a new room does not demonstrate anything, the authors have to rephrase to make their point clear.

**Details Of Ethics Concerns:**

nothing specific

---

> ### Author Response · Authors · 2024-12-03
> **Official Comment by Authors**
>
> We sincerely thank the reviewer for their time and effort in providing thoughtful and constructive feedback on our paper. We found the suggestions and comments to be helpful in reiterating our work. However, we were unable to fulfill all the requests effectively on time and therefore have decided to withdraw the paper. We feel that addressing all the points raised (including Problem formulation, comparison and benchmarking against GateL0RD, THICK WMs) requires making significant changes to the paper that might be incompatible with the current version.
>
> We are committed to thoroughly addressing your feedback and improving our presentation. Our intention is to resubmit the paper once it is more mature and would reflect the level of quality and rigor expected. Below, we have provided responses to some of the concerns raised in the review. While this may not address all the points, we hope it demonstrates our commitment to improving the manuscript:
>
> 1. GateL0RD [1] and THICK World Models [2]: We thank the reviewer for bringing this line of work to our attention. We will use them as our baselines.
> 2. Supervised contrastive learning is not the crux of our method. It was an addition that served to show that our model can learn human-aligned abstractions - if required, (Figure 5, 7) similar to how human brains adapt from their surroundings by categorizing observations. The highlight of our paper is Figure 3, representing the top-down modulation that allow abstractions of transition functions. We will make this more clear in our revision.
> 3. In Figures 5 and 6, the (x, y) positions do not mean anything significant and we have found them to converge to different (x, y) locations during repeated runs. However, the relative distance remains. The Habitat examples in Figure 4 and 7 show similar behavior, except that in Figure 4 without the use of Supervised Contrastive Learning, the abstractions are not tightly packed in 2D space. This could be attributed to the noise in the dynamics arising from high dimensional RGBD images.
> 4.  “replacing autoencoders … with ViT … is straightforward”: so why didn’t the authors do it? : Pretrained encoders on habitat were available as open-source model weights from [3]. We have mentioned this in Section 4.3. Since we are academic researchers with access to very few GPUs, we prioritized pretrained visual encoders that would otherwise require millions of frames and significant compute resources [3].
> 5. Why are the episodic rewards decreasing BEFORE the goal changed? Thank you for bringing this to our attention, we will correct the figure. We plot a running average of the rewards with a window of 150 steps, for clarity. The raw rewards are extremely noisy, as is the case for RL algorithms. The plots are left shifted by 150 for this reason. We have confirmed that the episodic rewards decrease only after changing the goal which is the expected behavior for an on-policy RL algorithm conditioned on a single goal. In fact, the 2nd goal change at 2000 episodes is closer to the initial goal which gives a small boost for the RL Agent.
>
> [1] Gumbsch Christian, et.al. (2021)
>
> [2] Gumbsch Christian, et. al. (2021)
>
> [3] Wijmans Erik, et.al. (2020)

---

### Official Review · Reviewer_doSy · 2024-11-03

**Soundness:** 2
**Presentation:** 2
**Contribution:** 3
**Rating:** 3
**Confidence:** 4

**Summary:**

The paper proposes a worldmodel architecture that uses hypermodels and runtime inference for learning abstractions. For realistic environments, additional supervision in terms of sparsely labeled location-specification was needed to obtain a reasonable abstract clustering. The system is tested in a toy gridworld and in Habitat 2.0. The model can be used for highlevel planning and reduces the planning overhead drastically.

**Strengths:**

- interesting and reasonable hierarchical model architecture
- analysis of the method on Habitat, so realistic environment
- good visualizations
- visualizations and insights into latent representations are given

**Weaknesses:**

-  the paper is not very well written (partially unfinished)
- no fair baselines, the method is not compared to Dreamer or TDMPC or THICK world-models ( https://openreview.net/forum?id=TjCDNssXKU )
- no action incorporation into the higher highlevel
- the name of the method is misleading: I see only one small evidence of compositionallity with the two trivial gridwords, but other than that the architecture has no particular bias towards creating compositional structures and I would expect a much stronger empirical evidence if you want to claim that compositionallity emerges.

It is unfortunate the paper is not really carefully edited before submission. There are some unfinished sentences and missing glue in the paper.
I like the overall approach, but from what is presented here, it does not seem ready yet. Fair comparisons and ablation studies are missing.

- many small details: Letters are reused or mixed up: Example $\tau$ is used for time scale, temperature and also on one case for the inner inference iterations, but called $K$ in the caption of Fig 3.

**Questions:**

- I did not understand exactly how the timescales interact. You use T for the high-level timescale, but it is not clear to me when this is updated. Do you make the inference only every $\tau$ steps are every step?

- which exact algorithms are used for Fig 9? What is the planning horizon for the planners?

- Fig 8: I would expect a comparison between the case with high-level model and without.

---

> ### Author Response · Authors · 2024-12-03
> **Official Comment by Authors**
>
> We sincerely thank the reviewer for their feedback on our paper. We were unable to perform the necessary comparisons and baselines, and have decided to withdraw the paper from consideration. Our intention is to resubmit an improved version of the paper that reflects the expected rigor and quality. We address the questions raised in the review:
>
> - Timescales: We will make this aspect more clear since it is a cause for confusion. We have two variables **time steps**($\tau$) and **timestamp**($t, T$).  T is used as the higher level **timestamp**: $s_{t, T}$ represents the state at $t^{th}$ lower level timestamp and $T^{th}$ higher level timestamp. Figure 2 helps make this clear. Suppose we have an agent taking an action in the environment every step, we denote current “lower-level” **timestamp** with $t$. However, the agent also maintains a “higher-level state” variable that updates every $\tau$ steps instead of updating every step (similar to options in RL). In our experiments (both gridworld and Habitat), we show results with $\tau =$ $10$ and $15$ (Figure 2, 5). This means that the higher-level variable (we use $s^{(2)}$ in the paper) updates every $10$ steps. This update is posed as an inference problem in the paper and hence, we say that the inference process runs for $\tau$ steps. We will make this more clear in the next version of the paper.
> - Planning Algorithm: The results are shown for the gridworld environment. For the RL baseline and the lower level Composer agent, we use off-the-shelf policy gradients with baselines, advantage estimates and gradient clipping. This approach is similar to PPO and allows for faster convergence. Planning is done via Model Predictive Control (Random Shooting). The lower level agent is given an oracle transition model and reward function which correctly predicts next states. The planning horizon for the lower level agent is 10. Composer uses the learnt higher level state model to plan, with a horizon of 3. We use $\tau = 10$ here, but even if $\tau=5$ was used, a horizon of 3 at the higher level would imply a lookahead of $5 \times 3 = 15$ steps. This is the advantage of using hierarchical planning.

---

### Official Review · Reviewer_o9fL · 2024-11-03

**Soundness:** 2
**Presentation:** 3
**Contribution:** 2
**Rating:** 3
**Confidence:** 4

**Summary:**

## Compositional World Models with Interpretable Abstractions
This paper introduces an instantiation of a world model that distinguishes between high-level and low-level dynamics. The introduced method Composer turns long horizon modeling tasks into a series of smaller tasks, guided by a high level abstraction. They generate hypernetworks, conditioned on a high level abstraction, that define low-level dynamics. Composer is tested on a variety of navigation tasks in toy domains like Gridworld and realistic domains like Habitat.

**Strengths:**

This paper is written clearly and motivates the need for abstracted world models. The usage of hypernetworks for this task is novel and an interesting approach to solving the hierarchical problem. The authors also provide interesting ablative experiments of their method in gridworld.

**Weaknesses:**

There are several key issues with this paper that prevent it from achieving a higher score.
1. **Contrastive loss as after an aside**: Section 2 describes the pipeline for learning abstract representations of state for the purpose of learning low-level dynamics. However, the losses in (7) and (9) alone might lead to a local optima where the abstract representation contains no information. Thus, the method relies on the contrastive loss in (11) to prevent collapse of the abstract representation. However, the paper is structured such that this nuance is lost and that the contrastive loss is used solely for scaling.
2. **experiments do not indicate that abstraction is necessary**: The main task that is used in the experiments is navigation. However, the underlying dynamics in experimental domains do not change from room to room. To my knowledge, the Habitat navigation tasks used in this paper do not significantly benefit from the abstraction described here. Mainly, language-navigation tasks (such as "Go pick up the tooth brush" do require abstraction because room abstractions help condition the nav policy. It it imperative that the authors clarify why their chosen navigation tasks do in fact require a significant amount of abstraction.
3. **Usage of neuroscience**: The ideas in the paper can stand alone without the motivation of mammalian neuroscience. I would suggest removing the neuroscience oriented text because it does not add to the content of this paper.
4. **Experiments do not include significant baselines**: The purpose of learning compositional world models is to learn policies that are more generalizable and efficient than other model-free or model-based methods. However, from the experiments, this is not clear. I can only garner qualitative attributes of the method from the experiments. It is important to compare Composer with other world model and model-free policy learning methods. As such, I suggest at least comparing against DD-PPO (a baseline already in Habitat-lab), DreamerV3 and TD-MPC2. Or equivalent baselines if these are unsuited to your tasks.

Minor:
Line 142 seems to have text not meant for the submitted manuscript.

**Questions:**

In addition to addressing my comments in the Weakness section I would like the following questions answered.

1. Is the contrastive loss necessary for learning diverse representations?
2. Would labeling with VLM,  as suggested in the paper, solve of the navigation tasks similarly to your abstracted method?

---

> ### Author Response · Authors · 2024-12-03
> **Official Comment by Authors**
>
> We sincerely thank the reviewer for their feedback on our paper. We were unable to perform the necessary comparisons and baselines with DD-PPO, Dreamerv3 and TD-MPC2, and have decided to withdraw the paper from consideration. Our intention is to resubmit an improved version of the paper that reflects the expected rigor and quality. We address the questions raised in the review:
>
> 1. Is the contrastive loss necessary for learning diverse representations? No, it is not necessary as shown in Figure 5. However, it depends on what kind of diversity is expected. Our original hierarchical model is meant to detect diverse dynamics without any supervision and contrastive loss. But in practical applications like navigation in a home environment (Habitat 2.0), the different dynamics might not be useful. For example, in Figure 4C, it is possible to color the plot in a way that makes the points separable in 2D. But these clusters might not make any sense in the environment considered. What is useful, is a cluster that we can make sense of - different rooms. It is not guaranteed that the rooms have different vision-based transition functions or similar functions $P(s_{t+1}|s_t, a_t)$ within the room. In these cases (ex: home robots), we can use contrastive loss to learn diverse representation.
> 2. Would labeling with VLM, as suggested in the paper, solve of the navigation tasks similarly to your abstracted method? Labeling with a VLM is complementary to our method. We can use the labels generated by a VLM to learn diverse transferable dynamics.

---

### Note · Authors · 2024-12-03

**Comment:**

We sincerely thank all the reviewers for their time and effort in reading our paper and giving critical feedback. Overall, we feel that addressing all the issues raised (formulation, comparison and benchmarking against Dreamerv3, TDMPC2, THICK WMs, DD-PPO) requires making significant changes to the structure of paper that might be incompatible with the current version. Hence, we wish to withdraw the paper: Our intention is to resubmit the paper once it is satisfactory and would reflect the level of quality and rigor expected by the ICLR community.

**Withdrawal Confirmation:**

I have read and agree with the venue's withdrawal policy on behalf of myself and my co-authors.